# HazeSpace2M: A Dataset for Haze Aware Single Image Dehazing

## ABSTRACT

Reducing atmospheric hazes and enhancing image clarity is crucial for a range of applications related to computer vision. The lack of real-life hazy ground truth images necessitates synthetic datasets, which often need more diverse haze types, impeding effective haze type classification and dehazing algorithm selection. This research introduces the HazeSpace2M dataset, a comprehensive collection of over 2 million images designed to enhance the performance of dehazing through haze-type classification. HazeSpace2M includes diverse scenes with 10 haze intensity levels, featuring Fog, Cloud, and a novel category, Environmental Haze (EH). Leveraging the dataset, we introduce a novel technique of haze-type classification followed by specialized dehazers to dehaze hazy images. Unlike the conventional methods, our approach classifies haze types before applying type-specific dehazing, improving clarity and functionality across applications lacking real-life hazy images. We benchmark the state-of-the-art classification models against different combinations of the hazy benchmarking datasets (HBDs) and the Real Hazy Testset (RHT) from the HazeSapce2M dataset. For instance, ResNet50 and AlexNet, on average, achieve 92.75% and 92.50% accuracy, respectively, against the existing synthetic HBDs. However, the same models furnish 80% and 70% accuracy, respectively, against our RHT, proving the challenging nature of our dataset. Additional experiments utilizing our proposed framework verify that haze-type classification followed by specialized dehazing enhances dehazing results by 2.41% in PSNR, 17.14% in SSIM, and 10.2% in MSE over general dehazers. These results highlight the significance of Haze-Sapce2M and the proposed framework in addressing the pervasive challenge of atmospheric haze in multimedia processing. The codes and dataset will be available on GitHub soon.

## CCS CONCEPTS

• **Computing methodologies** → **Reconstruction**.

## KEYWORDS

HazeSpace2M, Haze type classification, Haze aware dehazing, Single image dehazing, Haze classification, Atmospheric haze, Multimedia

## 1 INTRODUCTION

Atmospheric haze significantly compromises image clarity, posing difficulties for computer vision tasks in autonomous systems, remote sensing, and surveillance [64]. Adverse weather conditions

*ACM MM, 2024, Melbourne, Australia*
© 2024 Copyright held by the owner/author(s). Publication rights licensed to ACM.
ACM ISBN 978-x-xxxx-xxxx-x/YY/MM
https://doi.org/10.1145/nnnnnnn.nnnnnnn

that reduce visibility can lead to accidents, as documented in various studies [21, 30, 40, 56]. To tackle the issues of hazes, researchers have developed dehazing algorithms to counteract haze's effects on image quality [7]. Advancements in traffic systems and vehicle detection technology further necessitate enhanced visibility [13, 40, 52, 65]. Current efforts focus on refining models to restore clarity to images impaired by adverse environmental conditions [41, 58, 60]. However, there is a consensus on the necessity for versatile dehazing techniques across variable weather patterns [18], with a rich dataset being crucial for developing robust Convolutional Neural Network-based models for effective atmospheric dehazing.

Large datasets with varied scenes and haze types are scarce; the RESIDE SOTS [32] benchmark dataset covers synthetic hazy images but is limited to a single haze type. Similarly, the Cityscapes [11] dataset includes fog and rain conditions but is confined to street scenes, highlighting a deficit in comprehensive hazy image datasets. Current image restoration (IR) models often operate without recognizing the specific degradation type [31, 36, 42, 43, 57, 61, 63]. Although instruction-based IR methods [8, 10] improve performance by classifying degradation type, they rely on manual input, which is impractical for autonomous systems. An automated model that can identify and adapt to various haze types in the image is needed for effective dehazing without human intervention.

However, to train such versatile models, we need a dataset that offers various haze types across different scene types [18], which is absent in the literature. Identifying this gap in the literature, we develop a dataset named "HazeSpace2M," suitable for haze type classification and training haze type-specific specialized dehazers. We structure the "HazeSpace2M" dataset in a way that is suitable for haze type classification and training haze type-specific specialized dehazers. Leveraging this dataset in this paper, we also propose a novel idea of an intelligent image dehazing approach that performs specialized dehazing based on the haze type present in an input image. Thus, our research makes significant progress in the direction of image dehazing and classification, marked by the following contributions:

- **Development of a Benchmarking Dataset:** We developed HazeSpace2M as a comprehensive benchmarking dataset designed explicitly for haze-type classification in single input images. Additionally, we are the first to introduce a hazy dataset for different scene types, especially the Farmland scene type, which is unparalleled in the literature. This dataset surpasses existing datasets in terms of number of images (over 2 million), scene types, type of hazes, and haze intensity (10 levels).
- **Intelligent Haze Aware Dehazing:** We propose a novel framework that performs dehazing with specialized dehazers based on the haze type present in the input hazy image.
- **Benchmarking SOTA Models:** We evaluate leading classification models, setting new benchmarks for haze-type classification accuracy.

- **Evidence for Specialized Dehazers' Efficacy:** Our findings demonstrate that specialized dehazers, informed by accurate haze type classification, enhance dehazing performance, surpassing the capabilities of generalized dehazing models.

## 2 RELATED WORKS

In recent years, various hazy image datasets [4, 5, 11, 23, 32, 34, 46, 53, 54] have emerged to aid in developing single image dehazing techniques. These datasets offer a range of images affected by different hazes. For instance, the RESIDE dataset [32] encompasses a variety of images, including both indoor and outdoor settings, with hazy conditions and their corresponding ground truth (GT) images. However, it lacks distinct subsets for various types of images and haze conditions. Conversely, the Cityscapes dataset [11] provides fog and rain-afflicted street scenes but lacks variety in scene types. The synthetic image collections FRIDA [54] and FRIDA2 [53] are designed primarily for algorithm assessment in visibility and contrast restoration, encompassing 90 and 330 images across urban road scenes, respectively. Despite their utility, the synthetic nature of these sets limits their effectiveness in modeling the complexity of real-world hazes.

To bridge the above mentioned gaps, the NH Haze [6] dataset, introduced during the NTIRE2020 [1] challenge, features 55 outdoor scenes with actual haze conditions alongside their haze-free GT images, proving invaluable for developing new dehazing methods. Moreover, the Haze4k dataset [34]-split into 3,000 training and 1,000 testing images—provides ample data for benchmarking novel dehazing approaches. Adding diversity, the Kede [37] dataset contains 225 images with nine groups showcasing different outdoor settings and haze thicknesses. In contrast, the O-HAZE [5] dataset with 45 scenes captured under consistent lighting conditions offers realistic pairs of hazy and clear images, facilitating the study of dehazing in authentic environments. In the realm of remote sensing, datasets like Haze1k [23] and RS Haze [46] enrich the dehazing research by providing images categorized by haze density and showcasing a variety of cloud haze levels, respectively. Haze1k offers 900 images curated for remote sensing applications, whereas RS Haze challenges researchers with nine distinct haze levels in its 5,700 GT images. These datasets play a crucial role in enhancing the development of algorithms that deal with the nuances of hazy conditions observed in satellite imagery.

Overall, these datasets have become central to benchmarking the performance of single image dehazing techniques. Especially, the datasets like RESIDE [32] and Foggy Cityscapes [11] with their extensive collection, are excellent for generalizing models and have become a benchmark for assessing dehazing algorithms [9, 14–16, 22, 26, 27, 33, 39, 46, 47]. However, the ranges of haze types and scenes are limited in these datasets, scoping the improvement with more diverse datasets having various haze types for creating classification models capable of classifying various haze conditions. To fill this gap, we present a new dataset that is both broad and diverse, covering a wide range of scene types and haze types, paving the way for breakthroughs in the realm of single image dehazing in terms of developing robust haze type classification and dehazing algorithms.

**Table 1: Overview of HazeSpace2M dataset scene and haze types: annotated with Fog, EH, and Cloud, each with 10 distinct haze intensity levels.**

| HazeSpace2M | | | |
|---|---|---|---|
| Outdoor | Street | Farmland | Satellite |
| Fog | Fog | Fog | Cloud |
| EH | EH | EH | |
| 10 different levels of haze for each category | | | |

## 3 OUR DATASET: HAZESPACE2M

HazeSpace2M is a diverse and large dataset with over 2M images, including the GT and Hazy images of three different types of hazes: Fog, Environmental Haze (EH), and Cloud. To the best of our knowledge, we are the first to introduce both EH and Fog separately for scenarios such as Outdoor, Street, and Farmlands. HazeSpace2M is suitable for developing intelligent dehazing models based on haze-type classification.

Notably, the HazeSpace2M includes four main scene categories: Outdoor, Street, Farmland, and Satellite, encompassing three haze conditions: Fog, EH, and Cloud, as stated in Table 1. Each GT image from every scene type features ten corresponding hazy images, varying from low to high intense levels. The HazeSpace2M dataset, curated for research, includes an extensive collection of over 130,193 GT images and approximately two million hazy images, each categorized into distinct levels of haze intensity across various scene types. It also has a subset named Real Hazy Testset (RHT) that features 1,030 real hazy images for evaluating models. This comprehensive dataset not only paves the way for creating more robust dehazing models but also facilitates the development of algorithms capable of classifying the types of haze present, thereby contributing significantly to image processing and multimedia.

**Table 2: Sources and composition of GT in the HazeSpace2M dataset: a breakdown of the various image sources and the number of GT images selected from each source.**

| Scene Types | Image Sources | Total # of Images in Source | Total # of GT Images we Pick |
|---|---|---|---|
| Outdoor (OD) | ADE20K [66, 67] | 27,638 | 2,106 |
| | OTS [32] | 8,964 | 7,851 |
| | GSV [62] | 62,068 | 20,696 |
| | SFTGAN [59] | 10,200 | 4,596 |
| | Our Collections | 687 | 687 |
| Street (ST) | GSV [62] | 62,068 | 20,000 |
| | Cityscapes [11] | 19,998 | 19,998 |
| Farmland (FL) | Our Collections | 830 | 830 |
| Satellite (SL) | Haze1k [23] | 1,035 | 898 |
| | Forest Fires [17] | 42,815 | 42,815 |
| | DGLCC [12] | 1,146 | 1,146 |
| | DGRED [? ] | 8,570 | 8,570 |
| **Total GT Images:** | | | **130,193** |

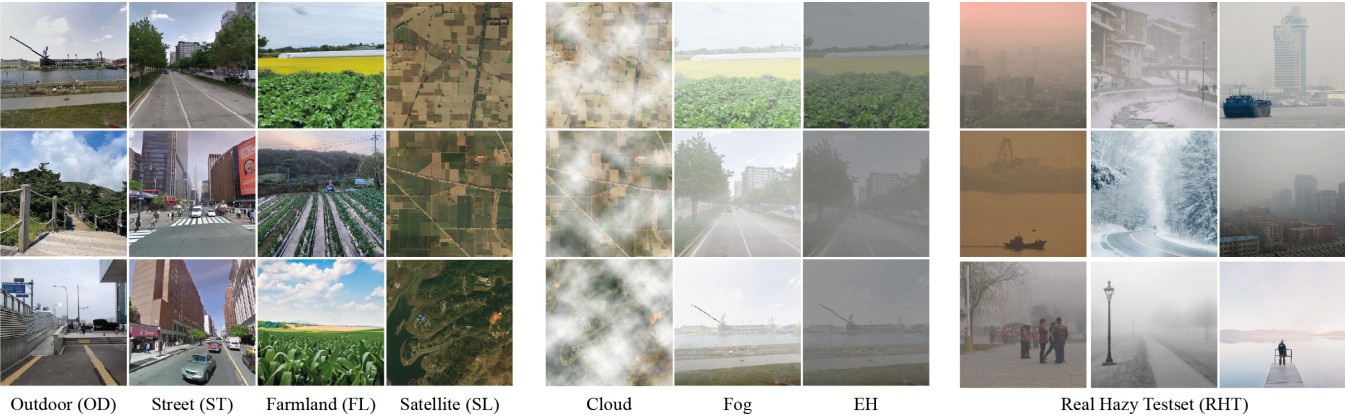

Outdoor (OD)     Street (ST)     Farmland (FL)     Satellite (SL)          Cloud          Fog          EH          Real Hazy Testset (RHT)

**Figure 1: HazeSpace2M at a glance: Showcasing Ground Truth (Green), Synthetic Hazy (Yellow), and Real Hazy (Red) images across diverse scenes.**

## 3.1 Data Collection and Generation

Before generating the hazy data, we collected a large amount of image data from various sources. These images are mainly the GT images in the HazeSpace2M dataset.

**Quality Assurance.** As shown in Table 2, we collect most of our GT images from the existing datasets or online under a Creative Commons License (CML), and some are the images captured from our personal devices. Cross-checking the quality of these images is a challenging but essential task. Initially, to ensure the quality of the GT images of our HazeSapce2M dataset, we established three conditions for excluding GT images while collecting from different sources. The conditions are as follows:

- Resolution: The image is of low quality.
- Haze Presence: The image contains haze in any form.
- Irrelevance: The image is not relevant to the scene types of HazeSapce2M.

If any image from our sources meets either of these criteria, it is excluded from the GT set of the HazeSpace2M dataset. For example, as shown in Table 2, we selected only 2,106 images from the ADE20k [66, 67] dataset out of 27,638 and 7,851 images out of 8,964 from the RESIDE SOTS [32] to use as GT images in the HazeSpace2M dataset. Similarly, we take 20,000 out of 62,068 images from the GSV [62] dataset as the rest match criteria 3. Thus, we ensure the quality and reusability of the GT images while we collect the GT images for HazeSapace2M from a wide range of sources.

**Scene Types.** As mentioned earlier, our HazeSpace2M dataset comprises diverse scenes. Outdoor images provide aerial and ground-level views of urban environments, capturing elements like architecture and traffic. Street view offers a closer look at urban roads and daily life. Farmland images focus on agricultural areas, detailing rural landscapes. Satellite images from high altitudes afford expansive views of the Earth's valuable surface for geographical and environmental studies and tracking changes in land use patterns, highlighting details unnoticeable at ground level. The images with the green line in Figure 1 display some sample images of different scenes of the HazeSapce2M dataset.

**Haze Types.** The HazeSpace2M dataset features three hazes types: Fog, EH, and Cloud. The haze types are applied to the GT images to create ten different haze intensities, which means that from each GT image, we produce ten hazy images of different haze intensity, which varies from light to dense.

**Fog:** Fog is caused by the presence of water droplets in the air, typically when there is a high relative humidity. It is a ground-level haze that reduces visibility.

**Cloud:** Cloud haze is characterized by cloud formations at various altitudes, affecting the lighting and contrast in images.

**Environmental Haze (EH):** EH is an atmospheric condition characterized by fine particles, aerosols, and pollutants suspended in the air. It is commonly caused by human activities, including industrial emissions and vehicle exhaust, but can also originate from natural sources such as burning from wildfires and agricultural lands. The images with the yellow line in Figure 1 display some sample images of different haze types of the HazeSapce2M dataset.

**Real Haze Testset (RHT):** The RHT comprises a collection of real-life hazy images sourced online to evaluate the ability of our classification models to identify haze types in real-world scenarios. These images are curated using specific search terms; for instance, searches for "foggy images," "foggy weather," and "winter fog" helped label images as Fog. Similarly, searches using "environmental haze," "air pollution," "wildfire," and "smoky environment" facilitated the labeling of images as EH. We meticulously verify each image's visual characteristics and origin to accurately represent the specified haze type. Thus, we collected around 686 images with fog haze and 344 with EH. The images with the red line in Figure 1 present some sample images of the RHT subset of the HazeSapce2M dataset. However, sourcing original satellite images depicting cloud haze posed a challenge. To address this, we incorporated 500 satellite cloudy images from the RS Haze [46] dataset into RHT, enabling comprehensive evaluation of the classification models trained on the HazeSpace2M dataset.

**Table 3: Details of the subdivision of the HazeSpace2M dataset according to scene types and haze conditions, listing the number of GT images and the generated hazy images across different subsets with the defined names for each subset.**

| Subset Names of HazeSpace2M depending on various Scene Types | Subsets Names of HazeSpace2M depending on various Haze Types | HazeSpace2M | | |
|---|---|---|---|---|
| | | Nature of the Image | # of GT Images | # of Hazy Images |
| Outdoor (OD) | Outdoor Fog (ODF)
Outdoor Environmental Haze (ODEH) | Synthetic
Synthetic | 35,936 | 359,360
359,360 |
| Street (ST) | Street Fog (STF)
Street Envirnmental Haze (STEH) | Synthetic
Synthetic | 39,998 | 399,980
399,980 |
| Farmlands (FL) | Farmland Fog (FLF)
Farmland Envirnmental Haze (FLEH) | Synthetic
Synthetic | 830 | 8,300
8,300 |
| Satellite (SL) | Satellite Cloud (SLC) | Satellite | 53,429 | 534,290 |
| **Real Haze Testset (RHT)** | - | Real | - | 1,030 |
| **Total:** | | | **130,193** | **2,070,600** |
| **Total # of Images (GT + Hazy) in HazeSpace2M dataset:** | | | **2,200,793 (2.2 Million Images)** | |

**Table 4: Comparative evaluation of image quality metrics across the existing datasets. The comparison of PSNR and SSIM values for the lowest and highest haze levels across different datasets, including our HazeSpace2M dataset.**

| Datasets | Scene Type | Haze types | | | # of GT Images | # of Hazy Images | Lowest Haze Level | | Highest Haze Level | |
|---|---|---|---|---|---|---|---|---|---|---|
| | | Fog | Cloud | EH | | | PSNR ↑ | SSIM ↑ | PSNR ↑ | SSIM ↑ |
| FRIDA [53, 54] | Outdoor | ✓ | ✗ | ✗ | 84 | 420 | 27.54 | 0.81 | 29.92 | 0.69 |
| Foggy Driving [11] | Street | ✓ | ✗ | ✗ | 10,425 | 10,425 | 28.50 | 0.88 | 27.70 | 0.68 |
| I-Haze [34] | Outdoor | Not Specified | | | 30 | 30 | 29.34 | 0.85 | 27.57 | 0.48 |
| O-Haze [46] | Satellite | Not Specified | | | 45 | 45 | 28.96 | 0.80 | 27.49 | 0.37 |
| SOTS [32] | Outdoor | Not Specified | | | 8,964 | 313,950 | 29.27 | **0.99** | 27.44 | 0.83 |
| NH Haze [6] | Outdoor | Non-Homogenous | | | 55 | 55 | 28.43 | 0.66 | 27.70 | **0.22** |
| Haze1k [23] | Satellite | ✗ | ✓ | ✗ | 1,035 | 1,035 | 28.51 | 0.91 | 27.49 | 0.23 |
| RS Haze [46] | Satellite | ✗ | ✓ | ✗ | 6,000 | 54,000 | 27.57 | 0.97 | 27.27 | 0.52 |
| **HazeSpace2M** | Outdoor | ✓ | ✗ | ✓ | 35,936 | **718,720** | 30.91 | 0.98 | 27.11 | 0.25 |
| | Street | ✓ | ✗ | ✓ | 39,998 | **799,960** | 31.91 | 0.98 | 27.36 | 0.39 |
| | **Farmland** | ✓ | ✗ | ✓ | 830 | 16,600 | 32.32 | 0.97 | **27.08** | 0.23 |
| | Satellite | ✗ | ✓ | ✗ | 53,429 | 534,290 | **34.61** | 0.98 | 27.49 | 0.23 |

## 3.2 Annotation Process and Tools

Inspired by [23] and [46], we utilized Adobe Photoshop 25.1 with its advanced ML-based Neural Filters (NFs) to generate hazy images for our HazeSpace2M dataset [2]. We crafted Photoshop actions, which automate editing tasks, to create varied haze levels [3]. This approach allowed the efficient processing of our extensive dataset, consisting of over two million images, generated over several months using three computers.

## 3.3 Quantitative Analysis

The HazeSpace2M dataset, as shown in Table 3, incorporates Fog and EH hazing on its Outdoor (OD), Street (ST), and Farmland (FL) subsets, while the Satellite (SL) subset is treated with Cloud haze, creating subsets designated as ODF, STF, FLF for Fog; ODEH, STEH, FLEH for EH; and SLC for Cloud haze. The OD, ST, FL, and SL subsets consist of synthetic hazy images alongside RFH and REH, which are real hazy images. Fog and EH haze types applied across ten intensity levels to the OD subset's 35,936 GT images result in 718,720 hazy images for OD, equally split between ODF and ODEH. The ST and FL subsets yield 816,560 hazy images from 40,828 GT images, and SL comprises 534,290 Cloud-hazy images from 53,429

GT images. Totaling around 130,193 GT images, the HazeSpace2M spans approximately 2.2 million hazy images when considering all three haze types and ten haze intensities per GT image, detailed in Table 3.

Compared to established datasets in literature [4–6, 11, 23, 32, 46, 53, 54], Table 4 presents the comparative PSNR and SSIM metrics. The HazeSpace2M dataset demonstrates high PSNR and SSIM at the lowest haze level, reflecting clear images under minimal hazing. At the highest haze level, these metrics show a marked reduction, illustrating the substantial impact of intense hazing. This variance signifies the dataset's wide range of haze intensities, providing a broader scope for analysis than previous datasets. The HazeSpace2M also exceeds others in image volume, offering an extensive array of GT and hazy images. Including the FL subset introduces a new scene type to the dataset, enriching the diversity and research applicability. Comparing the scene types and the haze types, it is evident that the HazeSapce2M consists of a diverse type of scene and haze compared to the existing haze image datasets. Each subset within HazeSpace2M contains GT and hazy images, establishing its superiority in dataset quantity and scene variety.

**Figure 2: Our proposed framework for specialized dehazer-based intelligent dehazing based on the haze type classification in image enhancement workflows, including (A) training classifiers to recognize haze types, (B) using the classifier to identify the type of haze in a single input image, (C) selecting the appropriate dehazer based on the haze classification, and (D) the final dehazing process to clear the image from atmospheric obscurations with the selected specialized dehazer.**

## 4 PROPOSED FRAMEWORK

Our proposed approach for intelligent dehazing based on haze type classification is illustrated in Figure 2. It has four main blocks, each with a particular task. As shown in Figure 2A, we use the dataset to train the SOTA classification models [19, 20, 24, 25, 28, 35, 38, 44, 45, 48–51] to sort out the models that could classify haze in the single-input image. Thus, we benchmark the SOTA models against the existing synthetic hazy benchmarking datasets and the Real Hazy Testset of the HazeSapce2M. Then, as illustrated in Figure 2B, we use the trained classifier for classifying the haze in a single input image. In this paper, we train the SOTA classification models on the HazeSapce2M dataset for haze-type classification. As in Figure 2C block, based on the classification result and output haze type, the model selects a suitable dehazer and performs the inference accordingly in the Inference Block, which is illustrated in Figure 2D. As with the classification models, we train three dehazing algorithms for three different hazes, namely Fog, EH, and Cloud, on our HazeSapce2M dataset. These dehazers are trained based on the modified ASM [55] in 100 epochs. Utilizing these three specialized dehazers, the complete framework we propose for specialized dehazers-based intelligent dehazing based on the haze type classification is depicted in Figure 2.

### 4.1 Experimental Setups

We conduct several experiments in line with the methodology illustrated in Figure 2. Our focus begins with training and evaluating classification models, followed by assessing generalized and specialized dehazers using the HazeSpace2M dataset.

**Haze Type Classification.** For training and validating the classification models, we take subsets from the HazeSpace2M dataset and split them as follows:

**Train and Validation Dataset**: We train our models using a subset of 15,000 images from the HazeSpace2M dataset, evenly divided among the three haze types, with 5,000 images for each category. We allocate 85% of these images for training (12,750) and the remaining 15% (2,250) for validation.

**Test Dataset:** We assess models on synthetic and real-life hazy images, creating different sets of testing datasets using the existing Hazy Benchmarking Datasets (HBDs) [5, 11, 23, 32, 46, 53, 54]. We also test the models against the RHT subset of the HazeSapce2M.

To ensure uniform training, all models used a batch size 32, a 0.001 learning rate, and a 512-pixel resolution. Following the footsteps of DTMIC [29], each model underwent a 50-epoch training with a 10-step patience early stopping technique.

**Single Image Dehazing.** We introduce two terms, namely Specialized Dehazer and Generalized Dehazer, and defined below to differentiate between the training processing for each.

*Specialized Dehazer (SD):* This term refers to a dehazing model explicitly trained on images of a particular type of haze. For instance, a model trained exclusively on foggy images to dehaze fog-related obscurities is considered an SD.

*Generalized Dehazer (GD):* In contrast, the Generalized Dehazing model is trained on a broader spectrum of hazy images. The GD model is not limited to a specific type of haze but is designed to handle various hazy conditions.

To conduct experiments with both SD and GD, we utilize the same dehazer architecture as depicted in Figure 2D. This architecture is developed based on the modified Atmospheric Scattering

**Table 5: Performance evaluation of the SOTA models using accuracy (ACC), precision (PRE), and recall (REC) against different combinations of the Hazy Benchmarking Datasets (HBDs), highlighting their effectiveness for haze type classification. These models are trained on the HazeSpace2M dataset.**

| Models | Different Combinations of Hazy Benchmarking Datasets for Haze Type Classification | | | | | | | | | | | | Average ACC |
| | Fog: FRIDA EH: O-Haze Cloud: Haze1k | | | Fog: Cityscapes EH: NH-Haze Cloud: Haze1k | | | Fog: Cityscapes EH: NH-Haze Cloud: RS-Haze | | | Fog: Cityscapes EH: O-Haze Cloud: Haze1k | | | |
| | ACC | PRE | REC | ACC | PRE | REC | ACC | PRE | REC | ACC | PRE | REC | |
|---|---|---|---|---|---|---|---|---|---|---|---|---|---|
| **AlexNet** | 0.96 | 0.96 | 0.96 | 0.95 | 0.95 | 0.95 | 0.83 | 0.91 | 0.83 | 0.96 | 0.96 | 0.96 | 92.50 |
| ConvNextLarge | 0.88 | 0.93 | 0.88 | 0.86 | 0.92 | 0.86 | 0.80 | 0.93 | 0.80 | 0.88 | 0.94 | 0.88 | 85.50 |
| DenseNet121 | **0.98** | 0.98 | 0.98 | 0.90 | 0.96 | 0.90 | 0.63 | 0.95 | 0.63 | 0.90 | 0.97 | 0.90 | 85.25 |
| DenseNet161 | 0.91 | 0.92 | 0.91 | 0.89 | 0.95 | 0.89 | 0.68 | 0.87 | 0.68 | 0.88 | 0.94 | 0.88 | 84.00 |
| DenseNet169 | 0.94 | 0.93 | 0.94 | 0.94 | 0.95 | 0.94 | 0.73 | 0.87 | 0.73 | 0.93 | 0.95 | 0.93 | 88.50 |
| DenseNet201 | 0.96 | 0.96 | 0.96 | **0.96** | 0.96 | 0.96 | 0.78 | 0.94 | 0.78 | **0.97** | 0.97 | 0.97 | 91.75 |
| EfficientNet_B0 | 0.88 | 0.95 | 0.88 | 0.85 | 0.95 | 0.85 | 0.63 | 0.92 | 0.63 | 0.85 | 0.96 | 0.85 | 80.25 |
| EfficientNetV2Large | 0.90 | 0.93 | 0.90 | 0.87 | 0.91 | 0.87 | 0.65 | 0.88 | 0.65 | 0.88 | 0.93 | 0.88 | 82.50 |
| GoogleNet | 0.86 | 0.86 | 0.86 | 0.88 | 0.89 | 0.88 | 0.74 | 0.86 | 0.74 | 0.89 | 0.90 | 0.89 | 84.25 |
| Inception_V3 | 0.78 | 0.86 | 0.78 | 0.79 | 0.89 | 0.79 | 0.68 | 0.90 | 0.68 | 0.80 | 0.91 | 0.80 | 76.25 |
| MNasNet | 0.94 | 0.95 | 0.94 | 0.82 | 0.94 | 0.82 | 0.52 | 0.93 | 0.52 | 0.82 | 0.95 | 0.82 | 77.50 |
| MobileNetV2 | 0.92 | 0.95 | 0.92 | 0.80 | 0.95 | 0.80 | 0.70 | 0.96 | 0.70 | 0.81 | 0.96 | 0.81 | 80.75 |
| MobileNetV3 | 0.76 | 0.94 | 0.76 | 0.51 | 0.92 | 0.51 | 0.43 | 0.95 | 0.43 | 0.51 | 0.94 | 0.51 | 55.25 |
| **ResNet50** | 0.96 | 0.96 | 0.96 | 0.95 | 0.94 | 0.95 | **0.84** | 0.92 | 0.84 | 0.96 | 0.96 | 0.96 | **92.75** |
| ResNet101 | **0.98** | 0.98 | 0.98 | 0.94 | 0.96 | 0.94 | 0.78 | 0.92 | 0.78 | 0.94 | 0.95 | 0.94 | 91.00 |
| ResNet152 | 0.97 | 0.97 | 0.97 | 0.94 | 0.96 | 0.94 | 0.76 | 0.93 | 0.76 | 0.94 | 0.96 | 0.94 | 90.25 |
| ShuffleNetV2 | 0.86 | 0.87 | 0.86 | 0.90 | 0.91 | 0.90 | 0.76 | 0.94 | 0.76 | 0.90 | 0.91 | 0.90 | 85.50 |
| SqueezeNet1 | 0.96 | 0.96 | 0.96 | 0.90 | 0.94 | 0.90 | 0.71 | 0.96 | 0.71 | 0.91 | 0.96 | 0.91 | 87.00 |
| VGG16 | 0.95 | 0.94 | 0.95 | 0.93 | 0.92 | 0.93 | **0.84** | 0.92 | 0.84 | 0.95 | 0.94 | 0.95 | 91.75 |

**Table 6: Evaluation of the SOTA models against the Real Hazy Testset (RHT) of the HazeSapce2M dataset using accuracy (ACC), precision (PRE), and recall (REC).**

| Models | ACC | PRE | REC |
|---|---|---|---|
| AlexNet | 0.70 | 0.71 | 0.70 |
| ConvNextLarge | 0.63 | 0.72 | 0.63 |
| DenseNet121 | 0.46 | 0.69 | 0.46 |
| DenseNet161 | 0.58 | 0.67 | 0.58 |
| DenseNet169 | 0.56 | 0.65 | 0.56 |
| DenseNet201 | 0.68 | 0.71 | 0.68 |
| EfficientNet_B0 | 0.49 | 0.64 | 0.49 |
| EfficientNetV2Large | 0.48 | 0.67 | 0.48 |
| GoogleNet | 0.66 | 0.68 | 0.66 |
| Inception_V3 | 0.54 | 0.63 | 0.54 |
| MNasNet | 0.45 | 0.68 | 0.45 |
| MobileNetV2 | 0.60 | 0.71 | 0.60 |
| MobileNetV3 | 0.60 | 0.71 | 0.60 |
| **ResNet50** | **0.80** | **0.78** | **0.80** |
| ResNet101 | 0.70 | 0.72 | 0.70 |
| ResNet152 | 0.63 | 0.71 | 0.63 |
| ShuffleNetV2 | 0.67 | 0.72 | 0.67 |
| SqueezeNet1 | 0.65 | 0.73 | 0.65 |
| VGG16 | 0.70 | 0.69 | 0.70 |

Model (ASM) [55] for removing haze in a single input image as follows:

$$I(x) = J(x) \times t(x) + A \times (1 - t(x)). \quad (1)$$

The term $K(x)$ represents a combined variable that encapsulates both $t(x)$ and $A$, while $I(x)$ signifies the observed image with haze. Here, $A$ is the global atmospheric light and $t(x)$ is the transmission map defined as:

$$t(x) = e^{-\beta d(x)}, \quad (2)$$

where $\beta$ is the scattering coefficient of the atmosphere, and $d(x)$ is the distance between the object and the camera. The modified version of Eq. (1) that is proposed for LDNet gives improved performance for removing haze from the images, which is verified through comprehensive inferences on different datasets [55]. Hence, for the experiments in our paper, we employ the modified version of the ASM model that is stated as follows:

$$J(x) = K(x) \times I(x) - K(x) + b_{\text{bias}}, \quad (3)$$

where the bias term is incorporated with a default value of 1 and the encapsulated values of $t(x)$ and $A$, which we define by $K(x)$, as:

$$K(x) = \frac{\frac{1}{t(x)} \times (I(x) - A) + (A - b_{\text{bias}})}{(I(x) - 1)}. \quad (4)$$

For the experiments of single image dehazing and to know if SD performs better than GD, we use LDNet [55] that is developed based on Eq. (3) with the same hyperparameter settings as the backbone of our dehazer algorithms and train them with our mentioned training datasets in different steps as:

- LDNet: Trained using the RESIDE [32] dataset, a common benchmark in dehazing research.

- GDNet: Trained on a composite dataset of 150,000 images comprising images affected by Fog, Cloud, and EH to create a generalized model.
- SDNets: Individually trained on distinct haze types. Initially, the model is trained exclusively on Fog type hazy images, followed by training on Cloud type, and finally on EH type hazy images, with each model saved after training.

With these setups of the dehazers mentioned above, we organize the training, validation, and test datasets in the following manner:

**Train and Validation Dataset:** For the SD models, each targeting a specific haze type (Fog, EH, and Cloud), we select 5,000 GT images and their 50,000 corresponding hazy images at ten distinct intensity levels from each class within the HazeSpace2M dataset. Consequently, we train each SD model using these 50,000 hazy images. In contrast, for the GD model, we amalgamate the 50,000 hazy images from each of the three classes, resulting in a comprehensive dataset of 150,000 hazy images encompassing all haze types. In both scenarios, we divide the dataset into training and validation sets with a 90/10 split ratio, ensuring a balanced model training and validation approach.

**Test Dataset:** To evaluate the performance of the SD and GD models for different types of hazes, we curated a test subset from the HazeSpace2M dataset with the images unseen to the model. This subset comprises 1,000 hazy images for each haze category, distributed across ten distinct intensity levels. This test dataset verifies the models' robustness and effectiveness in handling a broad spectrum of haze types and varying levels of haze intensity.

## 5 EXPERIMENTAL RESULTS

Our dual experiments, single haze type classification, and dehazing for a single input image demonstrate the HazeSpace2M dataset's versatility and wide-ranging applicability. The experimental results of both experiments are discussed in the following sections.

### 5.1 Results of Haze Type Classification

Our evaluation of SOTA classification models on both synthetic and real hazy images commenced with a training phase of 50 epochs, subsequently assessing performance on the HBDs and RHT datasets are presented in Table 5 and Table 6. Initial results highlighted the challenge within the RHT subset, as most models fell short of achieving 80% accuracy. Nonetheless, ResNet50 surpassed this benchmark, showcasing its potential for single image haze type classification despite being only trained for a short period and training with a subset of the HazeSapce2M dataset.

Expanding our investigation, as stated in Table 5, some of the SOTA models namely AlexNet, DenseNet201, ResNet50, ResNet101, ResNet152, and VGG16 give over 90% accuracy on average against the different combinations of HBDs, while models like Inception_V3, MNasNet, MobileNetV3 give 70% accuracy below on average. The bold values for each accuracy (ACC) column represent the top accuracy among all the accuracies while testing the models against the corresponding combinations of the HBDs, while the underlined values represent the second-highest accuracies. The average ACC column shows the average accuracy achieved by each model against the HDBs. Exploring this column, we find that the AlexNet achieves an accuracy of 92.50%, while ResNet50 outperforms the AlexNet

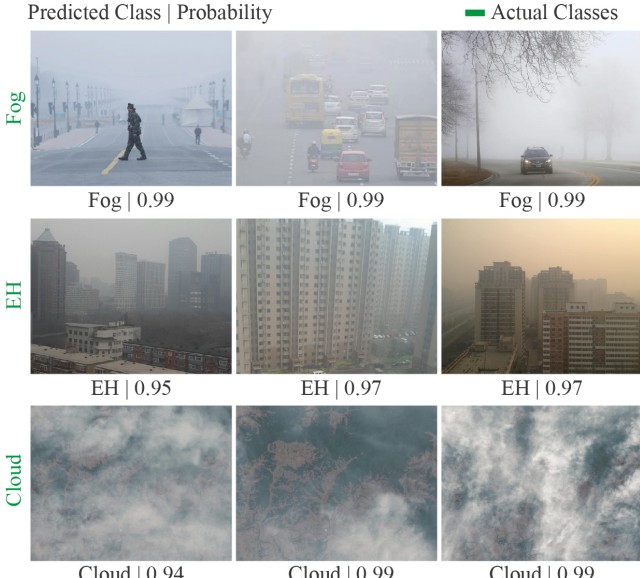

**Figure 3: Sample images from the Real Hazy Testset (RHT) for which the haze type is correctly classified by ResNet50, along with the prediction probabilities.**

with a slightly improved accuracy of 92.75%. Analyzing all these facts, we observed that ResNet50 and AlexNet performed robustly throughout the testing of different combinations of the HBDs.

We further evaluate all the models against the RHT to observe the performance of these models on images affected by the real atmospheric haze. As presented in Table 6, we still found ResNet50 to outperform the other models with an accuracy of 80%, while AlexNet, ResNet101, and VGG16 achieved 70% accuracy.

While several models result in very good accuracy on the synthetic datasets and very low accuracy on the RHT, the challenge lies in classifying haze types on a real hazy image. The ResNet50 shows some robustness by giving 80% accuracy, while the other models failed. The inference results on the RHT images are presented in the Figure 3, showing some of the correctly classified RHT images by the ResNet50 model. Overall, the results show the need to develop robust classification models that can outperform the existing SOTA models in the context of atmospheric haze-type classification.

### 5.2 Results of Single Image Dehazing

To investigate the effectiveness of SD models compared to GD models in single image dehazing, we conducted extensive evaluations using the HazeSpace2M dataset. Our methodology involved training the LDNet [55] model and its SD and GD variants across three distinct stages.

Our study rigorously evaluated the original LDNet dehazing model, achieving average PSNR, SSIM, and MSE values of 28.15, 0.65, and 99.89, respectively. We based these averages on comprehensive testing against various haze types, having 1000 hazy images for each haze type with detailed results outlined in Table 7.

Similarly, our evaluation of the GDNet and SDNet models on identical test sets, as detailed in Table 7, reveals that the SD models

**Table 7: Comparative performance metrics for LDNet, GDNet, and SDNets using PSNR, SSIM, and MSE scores for each model when subjected to dehazing tasks across various hazy conditions represented in fog, EH, and cloud test sets. The average scores reflect the overall performance of each model in processing unknown hazy images.**

| Testsets | LDNet | | | GDNet | | | SDNets | | |
|---|---|---|---|---|---|---|---|---|---|
| | PSNR ↑ | SSIM ↑ | MSE ↓ | PSNR ↑ | SSIM ↑ | MSE ↓ | PSNR ↑ | SSIM ↑ | MSE ↓ |
| **Fog Testset** | 28.47 | 0.78 | 92.46 | 28.47 | 0.77 | 92.31 | **28.55** | **0.85** | **90.49** |
| **EH Testset** | 27.89 | 0.44 | 105.44 | 27.93 | 0.63 | 104.78 | **28.34** | **0.79** | **98.43** |
| **Cloud Testset** | 28.11 | 0.75 | 101.76 | 28.29 | 0.70 | 97.85 | **29.84** | **0.83** | **76.17** |
| **Average** | 28.15 | 0.65 | 99.89 | 28.23 | 0.70 | 98.31 | **28.91 (2.41%+)** | **0.82 (17.14%+)** | **88.36 (10.12%+)** |

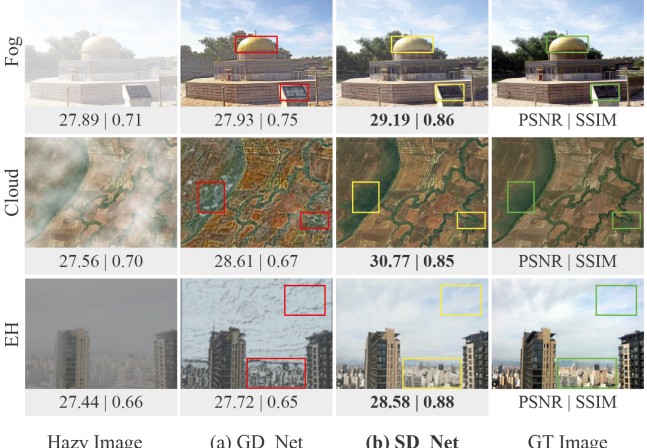

**Figure 4: Visual comparison of image dehazing results from LDNet, GDNet, and SDNet, with PSNR and SSIM metrics highlighted for clarity. Ground truth images are shown for reference, demonstrating the dehazing quality and the practical application of our proposed framework.**

surpass both the LDNet and GDNet in performance. The SDNet models demonstrate PSNR, SSIM, and MSE values of 28.91, 0.82, and 88.36, outperforming the GDNet, which records 28.23, 0.70, and 98.31.

The PSNR values show an improvement of around 2.7% for LD-Net compared to SDNets. Similarly, the performance of SDNets improved by 2.4% over GDNet considering the PSNR values from the experimental results presented in Table 7. Moreover, when we examine the SSIM and MSE values, we see a clear performance difference among these models. For example, the SDNets yield around 26.15% higher SSIM and 23.75% improved MSE than the original LD-Net. Similarly, compared to the GDNet, the SDNet models show an increase of around 17.14% in SSIM and 10.12% enhancement in MSE for dehazing images affected by Fog, EH, and Cloud. The SDNets outperform the other two models in all three metrics with a PSNR of 28.91, SSIM of 0.82, and MSE of 88.36, showing the effectiveness of an SD model over a GD model.

In addition to comparing performance metrics, the visual examination reveals the superiority of the SD model over the GD models. The single image dehazing examples in Figure 4 demonstrate the visual clarity achieved by the SD model is markedly better than

that produced by the original LDNet [55] and GDNet. Figure 4(c) presents the dehazed images of different haze types using the SDNet models. On the other hand, the inference results of the LDNet and GDNet, which we train traditionally with a relatively larger dataset, have been presented in Figure 4(a & b). To compare the visual clarity of the output images from each model, we highlight the differences via the rectangles. Considering PSNR and SSIM, the SDNet models give higher values than both traditional models, where LDNet has trained on RESIDE [32] dataset, and GDNet is trained on a subset of the HazeSpace2M dataset. We ran the inference on the images unknown to the models, i.e., we did not use these images to train the models. It should be noted that the training sets for the SDNet models are 50,000 images, whereas the training set for the GDNet model is 150,000. Even though we train the GDNet model with 3× more different haze types images, SDNets outperform GDNet, ensuring the superiority of our proposed framework.

This data conclusively supports the superiority of specialized dehazers-based dehazing techniques in enhancing single image dehazing. The clear implication is that classifying the type of haze in an image (as illustrated in Figure 2B) and subsequently applying the appropriate dehazing technique (Figure 2C) boosts the dehazing model's efficacy. Here, the Novel HazeSpace2M dataset leads the way by offering a diverse, large, and challenging hazy dataset with the confirmation of haze type classification, which results in better dehazing.

## 6 CONCLUSION

This paper introduces HazeSpace2M, an extensive dataset of over 2 million images designed to introduce haze-type classification and specialized dehazer-based image dehazing, addressing a critical need in computer vision for autonomous systems and security applications. While our computational resources limited training with the entire dataset, leading to reliance on partial datasets for model training, our results confirm HazeSpace2M's effectiveness in real-world haze condition classification, particularly highlighted by the RHT subset performance. Future efforts will focus on expanding the dataset's diversity in haze types, depths, and intensities and benchmarking dehazing models to fully leverage HazeSpace2M's potential. Our study demonstrates the significant role of accurate haze type classification in enhancing dehazing outcomes, offering a promising path forward for precision in image processing under adverse weather conditions, thereby filling a crucial gap in the field and setting the stage for future advancements.

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
