# OpenReview forum: "HazeSpace2M: A Dataset for Haze Aware Single Image Dehazing"
_acmmm.org/ACMMM/2024/Conference — MM2024 Poster_

### Official Review · Reviewer_scd8 · 2024-04-28

**Rating:** 2
**Confidence:** 4

**Summary:**

The study introduces HazeSpace2M, a comprehensive dataset of over 2 million images, designed to improve dehazing by providing diverse scenes with varied haze types, including a novel category called Environmental Haze. Utilizing this dataset, the researchers developed a method that classifies haze types before applying specialized dehazing techniques. This approach significantly enhances image clarity.

**Strengths:**

(1) The issue is worth researching in the image dehazing field, since there are few datasets for real-world dehazing.
(2) The authors make great efforts and contributions to proposing this large-scale dataset, which complements the scarcity in the current dehazing dataset.
(3) The proposed method can deal with diverse haze types.

**Limitations:**

(1) There are some typos in the article. For example, cite typos in Table 2 (DGREd).
(2) Although the proposed dataset is large and contains diverse types as claimed by the authors, it is still similar to the current synthetic dataset generated by ASM. In other words, the synthetic algorithm in PS still follows a certain distribution. Thus, I doubt the generalization of the model trained on the proposed dataset.
(3) The main concern is the proposed framework which applies multi-stage way. First, end-to-end model is the mainstream which is more suitable for real applications. Second, the classification accuracy is not so promising,  at least 95% accuracy should be achieved in my point cause it is the pre-requisite for dehazing. Lastly, the author not make comparisons in benchmarks. Although there are few benchmark for cloud and EH, there are lots of test sets for Fog. The author should prove their method can boost existing methods in these benchmarks.   If this point can be addressed in the rebuttal phase, I will consider to raise my scores.

**Suitability:**

2

---

### Official Review · Reviewer_hCdy · 2024-05-11

**Rating:** 4
**Confidence:** 4

**Summary:**

The authors propose a dataset of hazy images of size 2M called HazeSpace2M. This is a diverse set of images sourced from multiple different open source dataset + images collected directly by the authors. They have covered multiple different types of haze like cloud, fog, environmental haze etc. They also propose a method of dehazing based on first identifying the type of haze in the image and then used specialized algo based on that.

**Strengths:**

The size of the dataset (2Million) is a good addition for the open source community. The created dataset also has a good amount of variation and contains best picked data points from multiple open source datasets as well images collected by the authors.
The authors have done a thorough analysis by training multiple different models on the dataset and publishing results.

**Limitations:**

I am not so sure about the multi-step dehazing method proposed as the final results depends on the performance of both the first step prediction (haze type) and second step (actual dehazing) which gives multiple points of failure.

**Suitability:**

3

---

### Official Review · Reviewer_ds18 · 2024-05-25

**Rating:** 2
**Confidence:** 4

**Summary:**

This work introduces HazeSpace2M, a extensive dataset comprising over 2 million. The dataset features a diverse range of scenes, including outdoor, street, farmland, and satellite images, each subjected to different types and intensities of haze—fog, environmental haze (EH), and cloud. This level of diversity and size addresses a significant gap in existing datasets, which often lack the range of haze types and scene varieties needed for developing robust dehazing models.
A critical aspect of HazeSpace2M is its focus on haze-type classification, which is essential for selecting appropriate dehazing algorithms. This approach contrasts with conventional methods that apply general dehazing techniques without considering the unique characteristics of different haze types.

**Strengths:**

1. It is commendable to explore such a large-scale dataset.
2. With the evolution of existing dehazing methods, there is indeed a need for more up-to-date benchmarks.

**Limitations:**

1. I believe that mixing outdoor, street, farmland, and satellite images in a single dataset is highly inappropriate. Satellite images are vastly different from regular natural images, and their application scenarios are also significantly distinct. Under what circumstances would a system be required to perform outdoor dehazing and satellite image restoration simultaneously? I think the motivation behind this approach is highly unsuitable and does not align with the normal demands of the field.

2. The authors should focus on demonstrating how this dataset can further enhance the effectiveness of current dehazing methods or address some of the shortcomings of current dehazing methods, rather than concentrating primarily on haze-type classification. I think haze-type classification is not a problem worth striving for.

3. The authors should demonstrate that their method can improve the real-world generalization performance of dehazing methods.

**Suitability:**

2

---

### Meta-Review · Area_Chair_KVLo · 2024-07-04

**Recommendation:** Accept (Poster)
**Confidence:** 5

**Metareview:**

The authors' response addressed most of the reviewers' concerns and convinced them to raise their ratings. All three reviewers are recommending acceptance and the meta-reviewer agrees and invites the authors to further refine their work based on the received feedback and integrate contents from the response into their camera ready paper.